# Nucleocytoplasmic Transport: Regulatory Mechanisms and the Implications in Neurodegeneration

**DOI:** 10.3390/ijms22084165

**Published:** 2021-04-17

**Authors:** Baojin Ding, Masood Sepehrimanesh

**Affiliations:** Department of Biology, University of Louisiana at Lafayette, 410 East Saint Mary Boulevard, Lafayette, LA 70503, USA; masood.sepehrimanesh@louisiana.edu

**Keywords:** Alzheimer’s disease, amyotrophic lateral sclerosis, Huntington disease, neurodegenerative diseases, nuclear pore complex, nucleocytoplasmic transport, Ran GTPase

## Abstract

Nucleocytoplasmic transport (NCT) across the nuclear envelope is precisely regulated in eukaryotic cells, and it plays critical roles in maintenance of cellular homeostasis. Accumulating evidence has demonstrated that dysregulations of NCT are implicated in aging and age-related neurodegenerative diseases, including amyotrophic lateral sclerosis (ALS), frontotemporal dementia (FTD), Alzheimer’s disease (AD), and Huntington disease (HD). This is an emerging research field. The molecular mechanisms underlying impaired NCT and the pathogenesis leading to neurodegeneration are not clear. In this review, we comprehensively described the components of NCT machinery, including nuclear envelope (NE), nuclear pore complex (NPC), importins and exportins, RanGTPase and its regulators, and the regulatory mechanisms of nuclear transport of both protein and transcript cargos. Additionally, we discussed the possible molecular mechanisms of impaired NCT underlying aging and neurodegenerative diseases, such as ALS/FTD, HD, and AD.

## 1. Introduction

As a hallmark of eukaryotic cells, the genetic materials are separated from the cytoplasmic contents by a highly regulated membrane, called nuclear envelope (NE), which has two concentric bilayer membranes, the inner nuclear membrane (INM), and outer nuclear membrane (ONM). The ONM is continued by the endoplasmic reticulum (ER). There are some large specific protein structures across the NE, such as nuclear pore complex (NPC) that controls nucleocytoplasmic transport (NCT). NPC consists of special proteins, named nucleoporins (Nups), which play critical roles in regulation of the transport of most macromolecules back and forth across the NE [1]. NCT is a complex mechanism with involvement of many protein-protein interactions and recognitions, regulators and signaling pathways [2]. The transport across NE consists of import and export of protein and transcript cargos. All transcripts are transcribed in the nucleus and must be exported to the cytoplasm for protein synthesis, while certain proteins such as polymerases, histones, and transcription factors have to be imported into the nucleus to fulfill their functions [3,4]. Accumulating evidence revealed that impaired NCT is a fundamental pathological factor in aging [5] and aging-related neurological diseases [6,7], such as amyotrophic lateral sclerosis (ALS) [8,9], frontotemporal dementia (FTD) [10,11], Huntington’s disease (HD) [12,13], Alzheimer’s disease (AD) [14], Parkinson’s disease (PD) [15], Ataxia [16], and dystonia [17]. Understanding the regulation of NPC-mediated nuclear transport is critical to decipher the pathogenesis underlying impaired NCT and identify molecular targets for therapeutic interventions. In this paper, we comprehensively reviewed the components of NCT machinery and its regulatory mechanisms, including nuclear envelope and nuclear pore complex, importins and exportins, Ran GTPase and its regulators, and the regulations of nuclear transport of both protein and transcript cargos. Moreover, we discussed the possible mechanisms that link the impaired NCT to neurodegeneration.

## 2. Nucleocytoplasmic Transport (NCT)

### 2.1. Components of NCT Machinery

#### 2.1.1. Nuclear Pore Complex (NPC)

The NPC is the principal gateway between the nucleus and cytoplasm. There are more than a hundred to more than a thousand NPCs on the NE of a yeast cell [18] and a human cell [19], respectively. NPCs are huge protein complexes penetrating and bridging the inner and outer nuclear membrane [4,20]. A fully assembled NPC in vertebrates consists of multiple copies of about 30 different Nups with an estimated molecular mass of 120 MDa [1]. The three-dimensional structure of NPCs shows an eight-fold rotational symmetry and contains several major domains, such as cytoplasmic filaments, nuclear basket, central transport channel, and a core scaffold that supports the central channel [20,21]. The exact place of each Nup in an NPC plus their structure and functions are summarized in Figure 1 and Table 1.

Nups contain phenylalanine-glycine (FG) motifs or repeats which create a permeability barrier against passive diffusion of larger cargo molecules (>60 KDa) [22,23]. FG repeats are intrinsically disordered domains [24], and they directly function in mediating the passage of the soluble transport receptors pass through the NPC [2,25,26]. Despite of small molecules such as ions that pass the nuclear membrane via passive diffusion, all other macromolecules including RNAs and proteins need specific molecules called nuclear transport receptors (NTR), such as importins and exportins to achieve the specific nuclear trafficking [27,28].

#### 2.1.2. Importins and Exportins

Both importins and exportins belong to a protein family called Karyopherins (Kaps) [29]. These Kaps mediate transportation through binding to the nuclear localization sequence (NLS) or nuclear export sequence (NES) in the protein cargos and simultaneously bind to the FG repeats of Nups in the NPCs. In this way, Kaps also associate with Ras-related nuclear protein (Ran), a small regulatory Ras-related GTPase that controls both nuclear export and import (Figure 2). There are several human homologous Kaps with low sequence homology but similar biochemical properties such as molecular masses, isoelectric points, and architecture. Each of them specifically transports a set of proteins. Some of the Kaps just export or import the cargos and called exportins or importins, while some of them work as bidirectional transporter and called transportins. They recognize NES and/or NLS and help the protein and transcript cargos to pass across the NPC (Table 2).

### 2.2. NCT Regulations

#### 2.2.1. RanGTPase and Its Regulators

The NCT of most protein and transcript cargos (except mRNAs) depends on RanGTP-RanGDP gradient across the nuclear membrane, which provides a driving force and is generated by RanGTPase and its regulators [79]. The nucleus contains a high concentration of Ran that binds to GTP to form RanGTP, while the RanGDP is more concentrated in the cytoplasm. There is a mechanism for attaching/detaching of cargos from importins/exportins (Figure 2). Importins bind to the cargo in the cytoplasm in the presence of RanGDP and release cargo in the nucleus in the presence of RanGTP and exactly opposite manner occurs for exportins. In these processes, GTPase activity of Ran, the binding of Ran to GTP or GDP, and their subcellular distributions are tightly controlled by a set of regulators, such as Ran-binding proteins (RanBPs), Ran GTPase-activating protein (Ran-GAP), Ran guanine nucleotide exchange factor (Ran-GEF), and Ran-GDP dissociation inhibitor (Ran-GDI).

Ran has a low GTPase activity that is increased by attaching of other molecules including RanBPs and RanGAP. RanBPs work as scaffolding proteins that bind Ran and RanGAP. As RanBPs are anchored in the cytoplasmic side of the nuclear membrane, efficient conversion of RanGTP to RanGDP will occur only in the cytoplasm, resulting in a nuclear/cytoplasm ratio of RanGTP of approximately 200:1 [80]. Nuclear converting of RanGDP to RanGTP through regulator of chromosome condensation 1 (RCC1, also called Ran-GEF) further strengthens this RanGTP-RanGDP gradient [79,81]. Another regulator, Ran-GDIs, such as nuclear transport factor p10/NTF2 that inhibit the dissociation of GDP from Ran and keep Ran in an inactivated form of RanGDP until Ran-GEFs trigger the exchange of GDP to GTP [82] (Figure 2). After nuclear entering of importin α/β-attached cargo, binding of nuclear RanGTP to importin β causes nuclear releasing of the cargo, and then the complex of importin β/RanGTP will be dissociated and transported separately. With the help of RanBPs, RanGAP1 dissociates importin β from RanGTP by stimulating GTPase activity. During cargo export, the complex of exportin, RanGTP, and protein cargo is exported via NPC. RanGDP is transported from cytoplasm into the nucleus and converted to the RanGTP by the chromatin-bound Ran-GEF, forming the RanGTPase cycle [83] (Figure 2D).

NCT regulations consist of importing and exporting processes of protein and transcript cargos. On the protein side, they must contain a peptide signal known as NLS for importing to the nucleus and/or another specific signal called NES for exporting from the nucleus to the cytoplasm. NLS and NES signal peptides can be recognized by importins and exportins, respectively.

#### 2.2.2. Nuclear Import of Proteins with NLS

NLS is a sequence of positively charged amino acids including lysine and/or arginine. It usually localizes on the surface of the protein and can be recognized by importins (Table 2). In bipartite NLS, two positively charged amino acid sequences are separated by a spacer sequence and find in nucleoplasm, while monopartite just has one positively charged amino acid sequence such as SV40 T-antigen and c-Myc (Figure 3). Other types of discovered but uncommon NLS include mostly as proline-tyrosine amino acid pairing (PY-NLSs), and with lower frequency as the acidic M9 domain of hnRNP A1, the KIPIK sequence in yeast Matα2, a transcription repressor, and the complex signals of U snRNPs. Also, some specific NLS motifs are used to bind DNA and can be found near the DNA-binding region [84]. However, we can categorize common NLSs as two distinct types, basic classical-NLSs which described above and the PY-NLSs. Both types of NLSs must be recognized by importins. The heterodimer Impα-Impβ1 recognize most of the classical NLS. In molecular view, acidic part of Impα binds directly to the basic NLS side chains via electrostatic and polar interactions, its tryptophan residues hydrophobically interact with aliphatic parts, and its asparagine residues make hydrogen-bonds with the NLS main chains [85]. However, Transportin-1, which is a Kapβ2 protein, binds to PY-NLS [29]. This categorization of PY-NLSs is based on their loose N-terminal sequence motifs [86], which are either hydrophobic PY-NLSs containing hydrophobic residues of Glycine/Alanine/Serine or basic PY-NLSs enriched with basic residues. However, the c-terminal has a unique sequence as Arginine/Lysine/Histidine-sequence of 2 to 5 residues-Proline-Tyrosine motifs [87] (Figure 3).

In the process of protein import, protein cargos containing NLS are recognized by importins (such as Kapα and Kapβ1) in the cytoplasm and then bind RanGDP to form import complex (Figure 2). Importins will guide the complex to physically associate with Nups of NPC and then pass through the NPC with the driving force of RanGTP-RanGDP gradient [88,89]. In the nucleus, Ran-GEF exchanges the binding of GDP to GTP and disassembles the complex and releases the cargo in the nucleus. Importins will be exported to the cytoplasm and to be used for another run of protein import. RanGTP will participate the process of nuclear export [27].

#### 2.2.3. Nuclear Export of Proteins with NES

NES is a short signal peptide in the proteins which need to go out of the nucleus and contains four to five hydrophobic amino acids. This signal directs the protein out of the nucleus through the NPC by the help of nuclear transporter (Table 2). The diversity of NESs is very high. However, the typical NES has an L-X-X-X-L-X-X-L-X-L sequence in which L is a hydrophobic amino acid, often Leucine, but also can be Valine, Isoleucine, Phenylalanine, or Methionine, and X can be any other amino acids (Figure 3). The intervening amino acids routinely are negatively charged, polar, or small amino acids [90]. Any proteins that have this signal and also RNA molecules that bind to these proteins and form ribonucleoproteins (RNPs) can export from the nucleus (Figure 2D). Chromosome region maintenance 1 protein (CRM1) or Exportin-1 (XPO1) is the most important karyopherin that helps to export of most protein cargos with NES [91]. There are ten classes of NES including 1a, 1b, 1c, 1d, 2, 3, 1a-R, 1b-R, 1c-R, and 1d-R which bind to the hydrophobic pockets of CRM1 [92,93]. Most of NESs have low affinity for CRM1 and need another molecule with RanGTP-binding ability in the nucleus called RanBP3 [94]. There are no exact reasons for this low affinity, but it has been reported that bioengineered NES with high CRM1 affinity cannot pass through the NPCs due to firm binding to the NPCs [95].

In the process of protein export, exportin CRM1 binds protein cargos containing NES and RanGTP to assemble the export complex in the nucleus (Figure 2C). CRM1 mediates the nuclear export of the complex to the cytoplasm through NPCs. RanGTP-RanGDP gradient across the NE provides the driving force for the export. When the complex reaches the cytoplasmic side, RanGAP activates the RanGTPase activity and hydrolyzes the GTP to GDP to disassemble the complex and release the protein cargo into the cytoplasm [2,79]. Some RanBPs play critical roles in the export and the cargo release [96]. For example, Ran-Binding Protein 1 (RanBP1) induces a conformation change in the export complex, and promotes the disassembly of the complex and cargo release by increasing the RanGTPase hydrolysis that is activated by Ran-GAP1 [97,98]. RanBP2, also known as Nup358, is a large protein that constitutes the cytoplasmic filaments of NPCs [99]. It works as a scaffolding protein and provides a specific docking site for the nuclear export factor CRM1 [100]. RanBP2/Nup358 also exhibits a Small Ubiquitin-like Modifier (SUMO) E3 ligase activity [99,101], suggesting that sumolation at cytoplasmic filaments of the NPC may play an important role in the regulation of nuclear transport, at least for some substrates.

#### 2.2.4. Transcripts Export

The export of transfer (t) RNAs, micro (mi) RNAs, small nuclear (sn) RNAs and ribosomal (r) RNAs is also governed by the exportins of karyopherin family (Table 2) and the Ran-dependent NCT mechanism (Figure 2) [102]. However, the export of mRNA uses a non-karyopherin transport receptor and does not directly depend on the RanGTP-RanGDP gradient, making the mRNA export is mechanistically different from proteins and other RNAs [3,79] (Table 3 and Figure 4). Newly synthesized mRNAs are packed as large messenger ribonucleoprotein (mRNP) complexes, in which a single mRNA is associated with RNA-binding proteins (RBPs) that have functions in processing, capping, splicing and polyadenylation [21]. Thus, this exporting process involves three steps: (1) synthesis of pre-mRNA in the nucleus, processing, and packaging mRNP complexes; (2) targeting and translocation of mRNPs via NPCs or an NPC-independent mechanism called NE budding [103]; (3) intracytoplasmic release of the mRNPs for translation [104] (Figure 4).

The processing of nascent mRNA transcripts including 5′ capping [112,113], splicing [114], 3′ end cleavage, and polyadenylation [115] has direct influence on mRNA export. In human, there are about 30 heterogeneous nuclear ribonucleoproteins (hnRNPs), which bind to the nascent mRNA transcripts during transcription elongation and make mRNAs ready for further processing including packaging, export, and translation. Some of hnRNPs contain nuclear retention signals and should be removed before the export of mRNP. However, most of hnRNPs are attached to mRNA during export and come back to the nucleus after releasing in the cytoplasm [116].

In human, the NXF1-NXT1 (TAP-p15) heterodimer (1:1) functions as a general export receptor of mature mRNPs [117,118]. Like karyopherins, the NXF1-NXT1 complexes can physically interact with FG Nups and mediate RNP cargos pass through the nuclear membrane via NPCs. Like in the protein export process, RanBP2/Nup358 also provides a major binding site for NXF1-NXT1 dimers at the NPCs and functions in nuclear mRNA export [119]. However, once part of the mRNP reaches the cytoplasmic face of the NPC, the transport receptor NXF1-NXT1 heterodimer will be released in an ATP-dependent manner, rather than by GTP hydrolysis [120,121]. On the other hand, the mRNA export receptor NXF1 also plays important roles in coordinating transcriptional dynamics, 3′ end processing, and nuclear export of long three prime untranslated region (3′ UTR) transcripts [122]. The protein export receptor CRM1/Xpo1 also participates in the nuclear export of certain types of mRNAs, such as unspliced and partially spliced viral mRNAs [123]. Interestingly, besides the NPC-dependent pathway for nuclear mRNA export, another mechanism was identified in *Drosophila* body wall muscles called NE budding. In this mechanism, ultra-large ribonucleoprotein (megaRNP) granules containing mRNAs exit the nucleus by budding through the nuclear membranes independent of NPCs [103]. This NE budding pathway is proposed to be more prominent in particular biological processes or at certain growth stages that require high levels of protein synthesis, such as the rapid growth in early development or in response to stimuli [124,125].

Nuclear RNA export could be disrupted by any alterations in the exporting process, such as the changes in the NE including any alterations in the Nups which make NPCs, dysregulations of Ran gradient and its regulatory proteins, and variations in the proteins which attach to the RNAs and assemble RNPs [121,126]. Accumulating evidence indicates that RNA transport is impaired in several neurodegenerative diseases (NDs), which will be discussed in the following sections.

## 3. Impaired NCT in Neurodegenerative Diseases and Aging

### 3.1. Common Features of NDs

#### 3.1.1. Mislocalized and Aggregated Proteins Are Hallmarks in NDs

A typical pathological hallmark of NDs is the abnormal intracellular and/or extracellular protein accumulation in affected brain regions. It is believed that this dysregulation is directly involved in neurotoxicity, neurodegeneration, and finally causes clinical manifestation of the disease [127]. The intracellular inclusions detected in the brain of patients with a ND usually contain aggregates of mislocalized and misfolded disease-specific proteins. For example, aggregated β-amyloid peptide and intracellular neurofibrillary tangles (NFTs) containing aggregated and hyperphosphorylated tau protein are often found in Alzheimer’s disease (AD) [128,129]. TAR DNA-binding protein 43 (TDP-43)- or tau-positive inclusions can be detected in patients with ALS/FTD [130,131]. Intracellular Lewy bodies containing aggregated α-synuclein is a pathological feature in Parkinson’s disease (PD) [132]. The aggregates of superoxide dismutase (SOD) in motor neurons can be seen in ALS [133]. In Huntington’s disease (HD), the intranuclear inclusions of aggregated huntingtin protein containing polyglutamine (polyQ) expansion can be detected [134]. The exactly molecular mechanisms that trigger the formation of protein aggregates are not clear. Genetic mutations, environmental factors, and different stress conditions have been suggested to induce protein misfolding and aggregation in these diseases [127].

#### 3.1.2. Impaired NCT Is One Fundamental Pathogenesis of NDs

Accumulating evidence indicates that alterations of NCT are the fundamental pathological factors underlying these NDs (Table 4). Under normal NCT conditions (Figure 5A), the protein transport mechanisms enable each protein cargo to reach and retain in appropriate compartments, nucleoplasm, or cytoplasm. The right subcellular localization of proteins and the proper protein-protein interactions are foundations underlying their physiological functions. Similarly, in a healthy neuron, the majority of mRNAs should be exported to the cytoplasm for protein synthesis [79,135], including some RNPs that are delivered to axons or dendrites for localized translation, through which the cells are able to achieve the spatiotemporal regulation of gene expression with extraordinary precision [136]. The appropriate distributions of transcripts are extremely important in the regulation of gene expression and as well as in the maintenance of the hemostasis of local protein reservoir [137,138].

Dysregulations of protein NCT process will interfere with the normal distributions of protein cargos, leading to protein mislocalization (Figure 5B). Protein mislocalization could occur in nucleus or cytoplasm in diseased or aged neurons. Subsequently, the hemostasis of local protein reservoir could be disrupted, leading to abnormal protein-protein interactions, and triggering the formation of protein aggregates [176,177]. On the other hand, these aggregated proteins may have deleterious effects on neuronal functions by sequestering factors that are critical in signaling pathways or the NCT machinery. For example, by interfering with the NPCs, TDP-43 aggregates or poly-dipeptides encoded by C9orf72 repeat expansion in ALS/FTD and tau proteins in Alzheimer’s disease disrupt the NCT activities in diseased neurons [8,158,178]. These disruptions will further aggravate the impaired NCT activities and cause more severe protein and transcript mislocalization.

If nuclear transcript export mechanisms are impaired in a neuron (Figure 5C), the normal distribution of RNAs will be disrupted, such as nuclear RNA accumulation and mRNA mislocalization. These abnormal subcellular distribution of RNAs will dysregulate the protein synthesis, such as localized mRNA translation in axons and dendrites, leading to abnormal compartmental protein reservoirs [2,179,180]. Finally, the effects of impaired NCT, no matter the dysregulated protein transport or the disrupted nuclear RNA export, will be converged to neuronal toxicity and dysfunction, resulting in neurodegeneration and clinical manifestation of the disease (Figure 5D).

Neurons are specialized cellular subtypes that possess unique features and functions, including long processes and actively transport between soma and axon terminals, complex synaptogenesis, and synaptic connections. These features make neurons more vulnerable to the impairment of intracellular transport. The implication of defective NCT in NDs has been revealed by a variety of research model systems, including *Drosophila* [181,182], mouse [12,183], patient-derived induced pluripotent stem cells (iPSCs) [17,79,159,184,185,186], and autopsy brain of patients [187]. The reported NCT-defective diseases, related mutations and possible mechanisms were summarized in Table 4. In the following section of this review, we will take some diseases as examples to discuss how impaired NCT contributes to the pathogenesis of these diseases.

### 3.2. Impaired NCT in NDs

#### 3.2.1. Amyotrophic Lateral Sclerosis (ALS)

ALS is one of motor neuron diseases (MNDs) with excessive weight loss and muscles wasting even in the tongue, emotional lability, and cognitive dysfunction. Several pathophysiological mechanisms are involved in ALS including glutamate excitotoxicity, dysregulated interactions between neurons and glial cells, intracytoplasmic and intranuclear aggregation of certain proteins and RNAs, impairment of NCT and axonal transport, and changes in the axon terminals and neuromuscular junction (NMJ). Accumulating evidence indicates that defective NCT is a key mechanism in the pathophysiology of ALS. Sometimes, irregular nuclear morphology is seen in the ALS neurons. Irregularity of full lack of Nup62 has been reported in some cases of sporadic and familial ALS with SOD1 mutation [160] as well as mice model with mutations of SOD1 [188,189]. In addition, disruption of nuclear staining of Nup62 and Kapβ1 are reported in spinal motor neurons of sporadic ALS with clear mislocalization of TDP-43 [160]. Moreover, Roczniak-Ferguson and Ferguson reported a significant decrease in Nup188 protein level in human TDP-43 knockout cells which is contributed to the abnormal nuclear pore morphology [161]. Besides the dysregulations in NPCs and Nups, NCT defects are also related to the disruptions of Ran gradient, which provides the drive force for the nuclear transport of most transcript and protein cargos. For example, the hexanucleotide GGGGCC repeat RNA of C9orf72 could directly interact with RanGAP and cause it mislocalization [11,158]. By using fly model and the mammalian cells, Freibaum and coworkers reported that di-peptide repeat proteins (DPRs), which translated from six reading frames in either the sense or antisense direction of the hexanucleotide repeat [190], caused a loss of function in the Nup50 and changed the function of Nup153 and transportins, which help Nup50 and Ran in NCT [162]. Recently, Coyne and collaborators also found that GGGGCC repeat RNA of C9orf72 reduced eight Nups which initiated by reduction in POM121 and subsequently this reduction decreased the expression of other seven Nups and altogether affected the localization of Ran GTPase [191].

There are some controversies about the roles of these DPRs in the pathophysiology of ALS. Shi and collaborators reported that poly-PR dipeptides could disrupt NCT by attaching to the central channel of nuclear pores and locking the FG repeats of Nup54 and Nup98 [178]. In another study, Khosravi et al. reported that the non-coding region of C9orf72 gene repeat expansion related to poly-GA aggregations inhibited the nuclear import of a reporter containing NLS but did not affect a non-classical PY-NLS. In addition, they found that these poly-GA aggregations prohibited the nuclear transport of p65, which is induced by TNFα possibly through the impairment of importin-α/β-dependent pathway [163]. However, by using of Hela Kyoto cells that expressing the shuttling reporter NLSSV40-mNeonGreen2x-NESpki, Vanneste et al. demonstrated that poly-GR and poly-PR did not directly impair NCT. The exact molecular mechanisms underlying the impaired NCT by DPRs in ALS needs more investigations [192].

Mutations in the profilin1, a small actin-binding protein that regulates actin polymerization, cause NCT impairment in motor neurons in familial ALS. In the motor neurons with mutant profilin1, Ran, RanBP2 and RanGAP1 were misdistributed in the cytoplasm of instead of normal NE localization. Additionally, profilin1 mutations changed the nuclear membrane structure and clearly decreased nuclear import [9]. On the other hand, TDP-43 aggregates in cytoplasm contain certain NCT components including Nups, such as Nup35, Nup58, Nup62, Nup88, Nup93, Nup98, Nup107, Nup153, Nup155, Nup160, Nup205, Nup214, and Nup358, and transport factors, such as Xpo5, Gle1, Nxf1, RanGap1, and Ran [158,159]. Furthermore, in ALS with C9orf72 mutations, nuclear pore integrity could be rescued by modulation of actin polymerization, which plays an important role in connection of cytoskeleton and NCT [9]. Moreover, Bennett and collaborator found that sentaxin may play a role in the pathophysiology of ALS [193]. Sentaxin is a RNA-binding protein that regulates transcription but not its termination in higher eukaryotes [194]. They showed that TDP-43 is aggregated in the neuronal cytoplasm of mice with sentaxin mutations. Also, by immunocytochemistry for two NCT proteins including Ran and RanGAP1, they confirmed that the nuclear membrane was deformed and nuclear import was impaired in motor neurons of mutated mice [193]. Furthermore, a proline to serine conversion in the position of 56 in the vesicle-associated membrane protein-associated protein B and C (VAPB) has been reported in ALS patients, in which the transport of Nups and emerin to the NE was blocked and causes a NE defect [195]. Finally, as a lateral mechanism, it has been reported that motor neurons devoid of adenosine deaminase acting on RNA2 (ADAR2) are abnormally permeable to Ca^2+^ through the abnormal AMPA receptors. This intracellular increase of Ca^2+^ activates calpain, a Ca^2+^-dependent protease that cleaves TDP-43 into aggregation-prone fragments and also disrupts NCT by cleaving molecules involved in nuclear transport, including Nups [196]. Thus, nuclear envelope, NPC and Nups, Ran and Ran regulators, and other transport factors could be disrupted by mutations in ALS, leading to impaired NCT.

#### 3.2.2. Frontotemporal Dementia (FTD)

FTD is a clinical neurodegenerative disease and first described by Arnold Pick in 1892 [197]. It is characterized by progressive behavior deficits, executive function deficiency, language diminution, aphasia, lobar atrophy, and presenile dementia. FTD includes three clinical variants, behavioral variant FTD, non-fluent variant primary progressive aphasia, and semantic-variant primary progressive aphasia [198]. Less than 13% of patients with behavioral variant FTD and about 40% of patients with FTD show motor neuron disease [199]. In FTD, neuronal degeneration occurs mostly in the frontotemporal lobe and is distinguished by loss of neurons, gliosis, and certain alterations in the microvacuolar structure of frontal lobes [200]. The microtubule-associated protein tau (MAPT), the transactive response DNA-binding protein (TARDBP), and fused in sarcoma (FUS) are the most important affected proteins in FTD patients. Moreover, it has been reported that numerable cases of FTD with frontotemporal lobar degeneration had some inclusion bodies containing ubiquitin, neural precursor cell-expressed and developmentally down-regulated 8 (NEDD8) or p62, which are not FTD specific and can be seen in other ubiquitinated inclusions [200,201]. The most important mutations that seen in 35–65% of familial FTD patients include C9orf72, MAPT and GRN, while mutations in other genes are rarely seen [202]. Like ALS, a noncoding hexanucleotide repeat expansion in C9orf72 is reported as the most common genetic cause of FTD [203,204]. It is believed that the toxic gain of functions including translational repression, dysfunctions in mitochondria and nucleolus, and NCT disruption through the expression of DRPs are the main pathophysiological mechanisms [187,205].

NCT defects in FTD by DPRs occur through several mechanisms. Similar to ALS, DPRs disturbed the activity of Nup50, Nup153, and transportins as essential proteins in nuclear import [162]. In addition, the hexanucleotide repeat RNA of C9orf72 mislocalized RanGAP and disrupted its activity in the NCT [11]. Moreover, DPRs are neurotoxic and can cause neurodegeneration in cell culture and in animal models. By two unbiased genetic screens in a yeast model, overexpression of certain karyopherins including MTR10 and KAP104 did not change the mislocalization of DPRs in the rescue of DPRs toxicity. Also, in the 70–80% of induced neurons from patients with C9orf72, mislocalization of RCC1, also known as RanGEF, was occurred and just a very weak or no nuclear RCC1 signal was detected [10]. The MAPT which is encoded by MAPT gene on chromosome 17q21, stimulates tubulin polymerization into microtubules and stabilizes microtubules [206]. By using a model of MAPT mutation-induced FTD in human stem cells, Paonessa and colleagues found that cell body and dendrite mislocalization and hyperphosphorylation of tau in cortical neurons. This mislocalization apparently misshapes the nuclear membrane and directly influenced NCT which could be reversed by microtubule depolymerization [207].

#### 3.2.3. Huntington’s Disease (HD)

HD is a progressive neurodegenerative disease caused by a trinucleotide (CAG) expansion in exon 1 of the Huntingtin (HTT) gene, resulting in the expression of an expanded, mutant huntingtin protein (mHTT) [168]. In this disease, HTT protein has a long polyglutamine tract in the N-terminal. The most common cytological finding in HD is the aggregation of cleaved HTT protein in nucleus, cytoplasm, dendrites, and axon terminals. These aggregates are toxic and cause neuron death, mostly occur in striatal neurons in the basal ganglia [134]. The possibilities of selective neuronal degeneration in the striatal neurons could be due to the lack of brain-derived neurotrophic factor (BDNF) support, glutamate excitotoxicity [208,209], and expression of Rhes protein as a mediator of mHTT cytotoxicity in the striatal area [210].

Accumulating evidence indicates that the impaired NCT is another pathogenic factor of HD and mHTT could disrupt NCT by interfering with NPCs, Nups and other transport factors. For example, proteomic investigations confirmed that wild type HTT preferably bind to the nuclear import receptors importin-β1, 4, 7, and 9, while mHTT interacts with RanGAP1, the mRNA export factor Rae1, and the Sec13, revealing the abnormal protein-protein interactions between mHTT and NCT factors [169]. Nuclear accumulations of polyA-mRNA were seen in certain HD animal models and human patients [12,177], suggesting that the nuclear mRNA export was impaired in HD. Grima and collaborators evaluated different models of HD, including fly, mice, mHTT transgenic neuron, HD iPSC-derived neuron, and cadaver brain tissues, and found that mislocalization and aggregation of Nups and the NCT defects in these models. In their mouse model with the expression of exon 1 of human HTT with 125–160 CAG repeats, nuclear co-aggregation of mHTT with RanGAP1 and Nup62 was detected. They further demonstrated that the FG repeats in Nup62 and RanGAP1 attach to the mHTT [13]. Another study indicated that these inclusion bodies also contain Nup88 and Gle1, which are important components for mRNA export [12]. Consistently, the nuclear level of RanGAP1 decreased over the time in the disease and its diminishing was recovered by anti PIAS1 miRNA that decreased the mHTT level [13]. These findings suggested that mHTT comprehensively influence the components of NTC machinery.

Woerner and collaborators reported that just cytoplasmic aggregation, not nuclear and perinuclear inclusions, interfered with NCT of proteins and RNAs [177]. This is consistent with the finding that specific inhibition of CRM1 which leads to inhibition of nuclear export is neuroprotective and reverse NCT defect [13], suggesting that the toxicity of mHTT mainly occur in cytoplasm. The researchers also evaluated another heterogeneous mouse model that expresses human HTT exon 1 sequence containing 193 CAG repeat instead of the mouse HTT exon 1 and found that co-aggregation and localization of mHTT with RanGAP1 and Nup88 [13]. They also confirmed that the aggregation and mislocalization of RanGAP1 and cytoplasmic and intranuclear mislocalization of Nup62 also occur in the brain tissue of HD patients. More evidence of impaired NCT can be noticed in HD iPSC-derived neurons, including a significant decrease in the nuclear to cytoplasmic ratio of endogenous Ran, and an increase of the nuclear concentration of microtubule-associated protein 2 (MAP2) that normally localize in the cytoplasm, and the decrease of both RanGAP1 and Nup62 expression [13]. Moreover, a rodent cortical neuron transfected with HTT 82Q showed a clear cytoplasmic mislocalization of Ran that indicated the impairment of NCT in HD [13].

HD repeat-associated non-AUG translation (HD-RAN) proteins are novel homopolymeric expansion of poly(Ala) and poly(Ser) from the sense transcript, and poly(Leu) and poly(Cys) from the antisense transcript [211]. Despite repeat polyglutamine, the neuronal accumulation and toxic effects of HD-RAN proteins in the HD human autopsy brains were reported [212]. These HD-RAN proteins can also impair both active and passive NCT [13]. Furthermore, modification of Nup62 via O-linked N-acetylglucosamine (O-GlcNAc) transferase could be related to the HD pathology because of using an inhibitor of O-GlcNAc transferase in the primary mouse mHTT-expressing cortical neurons can rescue the NCT defects [213].

#### 3.2.4. Alzheimer’s Disease (AD)

AD is one of the important and most concerning neurodegenerative diseases worldwide especially in the elderly and contributes an estimation of 60–80% of dementia cases. It has been predicted that the number of people with AD grows to 13.8 million in the United States by 2050 [214]. The molecular pathophysiology of AD is closely related to the metabolism of the amyloid beta precursor protein (APP), a transmembrane protein that can be cleaved by α, β, and γ secretases. Just sequential cleavages by β secretase and γ secretase create two amyloid proteins of 40 and 42 amino acids. Aβ 42 aggregates as fibrillary amyloid protein in meningeal, cerebral vessels, and gray matter. This Aβ aggregation results in phosphorylation and aggregation of tau, a protein that stabilizes axonal microtubules, as twisted paired helical filaments of NFTs. These two aggregations cause neuronal degeneration and neuronal death in AD. On the genetic side of view, it has been reported that mutations in the APP gene on the chromosome 21, Presenilin1 (PSEN1) on the chromosome 14 and PSEN2 on the chromosome 1 that regulate the activity of γ secretase, and sortilin (SORT1) on the chromosome 1 that mediates surface APP transport to intracellular Golgi-ER complex are associated with AD. Additionally, an SNP (rs5848) in the GRN gene that leads to reduced PGRN levels and activates neuroinflammatory, pathological, and cognitive-based disease, is linked to the increased risk of AD [215]. Recently, Eftekharzadeh et al. reported that somatodendritic accumulation of tau enhanced the perinuclear tau concentration that consequently interacted with certain Nups including Nup98 and Nup62. These interactions resulted in the missorting of Nups into the cytosol and led to impaired NCT. They also found that nuclear leak and Ran mislocalization occurred in relation to the phospho-tau accumulation, which could ruin both active neuronal nuclear transport [14]. In AD, NFTs cause nuclear irregularity that can be seen simultaneously with irregular nuclear distribution of Nup62 and cytoplasmic aggregations of NTF2 [128]. These findings suggested that nuclear pores and Nups are disrupted in AD.

More evidence also indicated that other regulatory factors of NCT machinery were dysregulated in AD. Neuronal TDP-43 positive inclusions were detected in near 26% of confirmed AD patients [216]. Hippocampus of AD patients showed Hirano bodies, the intracytoplasmic inclusions that have been shown to contain importin-α1 [128,174]. NRF2, the nuclear transport of which is mediated by importin-α5 and β1, normally found in both cytoplasm and nucleus, but it is accumulated in cytoplasm in AD patients [217]. In AD mice model, importin-α1 expression was reduced and nuclear ultra-architecture continuum was disturbed [218]. Despite Nups and transportins, disrupted nuclear lamina [219] and nuclear localization of phospho MAPK/ERK kinase 1 in AD cases has been reported [220]. In both animal model and human brain, tau accumulation can reduce lamina protein levels and cause nucleoplasmic reticulum expansion, resulting in a nuclear invagination [219]. On the other hand, affected neurons in AD with neurofibrillary tangle showed proximity of paired helical filaments to the NE and NPC [221,222]. By using the transgenic mouse model, Ke and colleagues found that tau can deplete splicing factor proline- and glutamine-rich (SFPQ) from nucleus and induce its accumulation in cytoplasm of the amygdala [223]. Moreover, by evaluation of brain biopsies of patients with early and late stages of AD using electron microscope, it has been found that fascicles of paired helical filaments oppose the NE, NPC and the perinuclear polysomes and disrupted the crosstalk between nucleus and cytoplasm [221]. All these disrupted factors and pathways could contribute to the AD pathogenesis via the dysregulations of NCT.

### 3.3. Impaired NCT in Aging

Aging is an irreversible physiological process that is characterized by the progressive alterations in the metabolism of cells, impaired self-regulation, degeneration, and structural and functional changes [224]. Aging itself is the primary risk factor for most neurodegenerative diseases, including ALS/FTD, HD and AD that have been discussed above. The molecular mechanisms underlying the aging process and lead to NDs are still mysterious. It has been shown that the nuclear transport activities were compromised during aging due to dysregulations of NPCs, Nups and other transport factors, such as Ran-binding proteins [5,187]. Oxidative damage of long-lived scaffold Nups during aging is associated with defects in nuclear transport and the breakdown of the nuclear permeability barrier [225], suggesting the correlation between impaired NCT and aging process. This notion was further supported by the identification of long-lived proteins that include Nups [226]. Toyama et al. have conducted a system-wide identification of proteins with exceptional lifespans in the rat brain. They found that NPCs are maintained over a cell’s life through slow but finite exchange of even its most stable subcomplexes. This maintenance is limited and these proteins are inefficiently replenished after damage or protein levels decrease during aging, providing a rationale for age-dependent deterioration of NPC function and NCT [226]. Accelerated leakiness during aging and oxidative damages of Nups in old cells were reported. Nup93, as the establisher of the NPC diffusion barrier, was lost in old *C. elegans* cells but no changes were seen in Nup107 and Pom121 [225]. Aged-human fibroblast-derived neurons exhibited an age-dependent decline in the nuclear transport receptor RanBP17 that led to a loss of nucleocytoplasmic compartmentalization [5]. Furthermore, age-dependent decrease in the NPC density of dentate gyrus neurons in the rat brain was confirmed by electron microscopy [227]. Even if the number of NPCs stays constant like in the CA1 pyramidal cells of aged rat brain, accumulated NPC damage may still cause a lack of performing their functions [225]. Consistently, diffused protein aggregations were seen in aged *C. elegans* neurons [228], because Nups play a critical role in the prevention of protein aggregation in the neuronal cells [229]. Additionally, HeLa cells contained Hutchinson Gilford progeria syndrome (HGPS) mutation showed mislocalization of Nup62 and Nup153, decreased the lamina dynamics, and the impaired nuclear import [230]. Other studies include an age-related diminish in NCT [12] and age-induced accumulation of misfolded proteins [177] that caused NCT defect. These findings further demonstrated that impaired NCT may constitute a major molecular mechanism underlying aging process.

Indeed, NCT decreased and simultaneously protein aggregation increased with aging [231]. Oxidative damage is one of the possible links between aging and NPC defects. In young cells, Lamin B1 recruits the antioxidant transcription factor OCT1 and protects cells against oxidative stress [232]. Age-dependent down-regulation of lamin B1 increases cellular sensitivity to oxidative stress [233]. In aged neurons, this oxidative stress damaged NPC [225] and disrupted NCT [234]. Reactive nitrogen species (RNS) induce nitrosative stress that is associated with age-related neurodegeneration. For instance, the S-nitrosylation of CDK5 by nitric oxide stimulates nuclear lamina dispersion [235,236]. On the other hand, accumulation of a truncated lamin A mutant, progerin, caused impairment of nuclear import and decreased nuclear sumoylation due to oxidative or nitrosative stresses [237]. Nitric oxide can inactivate CRM1 via S-nitrosylation and impair CRM1-Ran-cargo export [238]. It will be interesting to further evaluate the association of nitrosative stress and affected NCT in neurons. Although the recent studies have greatly advanced our understanding of the aging biology, extensive studies are still required to reveal the exact pathophysiological mechanisms underlying impaired NCT in aging process, especially to dissect the causes and consequences between impaired NCT and aging.

## 4. Conclusions

Here we highlighted the major components of nucleocytoplasmic transport, the regulatory mechanisms and its linkage to neurodegenerative diseases and aging. Despite the identified impairments of NCT in different neurodegenerative diseases and aging, the exact molecular pathological mechanisms underlying these impaired NCT and the contributions to the clinical syndromes are not fully understood. In this emerging research field, at least the following directions should be emphasized in the future research. First, the regulatory mechanisms of NCT under physiological condition. The elucidation of detailed molecular mechanisms of NCT regulation is the prerequisite to understand the impaired NCT under diseased conditions, including the specific signaling pathways in the regulation of different cargos. Second, to develop reliable and efficient methods to screen NCT-defective conditions. In cultured cells, florescent reporters fused with NLS or NES are widely used for protein transport assay, and fluorescence in situ hybridization (FISH) is usually employed to measure the nuclear transcripts export [5,17,79]. However, these approaches work poorly at specificity and efficiency in brain tissues. Some biomarkers need to be identified and developed to reveal defective NCT activities in clinical samples. Third, to decipher the pathogenesis of defective NCT underlying diseased neurons and identify the molecular targets for therapeutic interventions. These molecular targets could be fundamental factors that regulate NCT process or some factors or pathways that are specifically involved in certain diseased conditions. The progress of neurodegeneration could be decelerated or even ameliorated if the intervention approaches are able to effectively rescue the defective NCT activities.

## Figures and Tables

**Figure 1 ijms-22-04165-f001:**
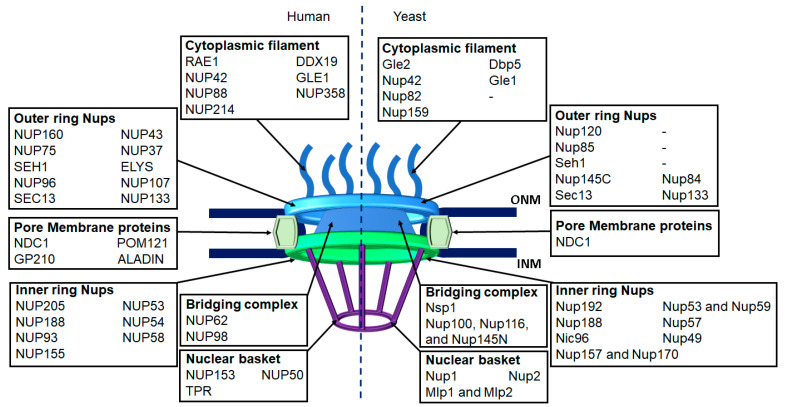
Nuclear pore complex structure and nucleoporins. Nuclear pores are embedded in the nuclear envelope and across the inner nuclear membrane (INM) and outer nuclear membrane (ONM). Each nuclear pore consists of outer ring, inner ring, pore membrane proteins, cytoplasmic filament, bridging complex and nuclear basket. Each part contains multiple copies of different nuclear pore subunits (nucleoporins). See Table 1 for more information about different nucleoporins.

**Figure 2 ijms-22-04165-f002:**
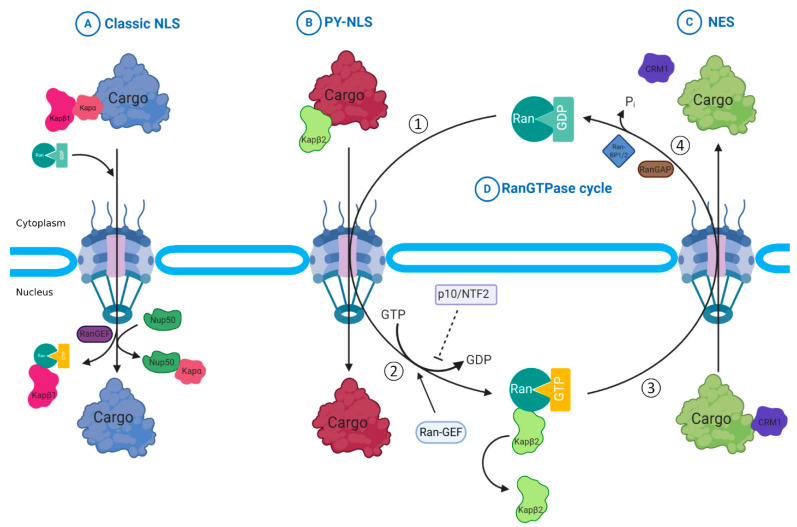
A Ran-dependent mechanism of protein nuclear transport. (**A**) Nuclear import of protein cargos containing classic nuclear localization sequence (NLS) is mediated by Kapα and Kapβ1. Import complex contains cargo-NLS, importin and RanGDP is formed in cytoplasm and imported into nucleus through a nuclear pore complex (NPC). The cargo is released from the complex after RanGEF exchanges the binding of RanGDP to RanGTP. (**B**) Nuclear import of cargos with the signal of proline-tyrosine amino acid pairing (PY-NLSs) is mediated by Kapβ2 in a similar mechanism as A. (**C**) CRM1 mediates the nuclear export of protein cargos containing nuclear export sequence (NES). Export complex contains cargo-NES, exportin (CRM1) and RanGTP is formed in the nucleus and exported into cytoplasm through a NPC. The cargo is released from the complex after the hydrolysis of GTP to GDP by RanGTPase, which is stimulated by RanGAP (Ran GTPase-activating protein). (**D**) RanGTPase cycle. ① RanGDP binds import complex in cytoplasm and it will be transported into nucleus. ② Ran-GEF exchanges the GDP to GTP and disassembles the complex in nucleus, resulting in the release of imported cargos. ③ RanGTP binds the export complex in the nucleus and it will be exported to cytoplasm. ④ RanGAP activates the GTPase activity of Ran and hydrolyze GTP to GDP, resulting in the release of exported cargos. In this cycle, another regulator Ran-GPI (p10/NTF2) could keep Ran at an inactivated form (RanGDP) by inhibiting the dissociation of GDP from Ran. Ran and Ran regulatory proteins generate and maintain the RanGTP-RanGDP gradient across the nuclear envelope and provide the driving force for the nuclear transport of protein cargos.

**Figure 3 ijms-22-04165-f003:**
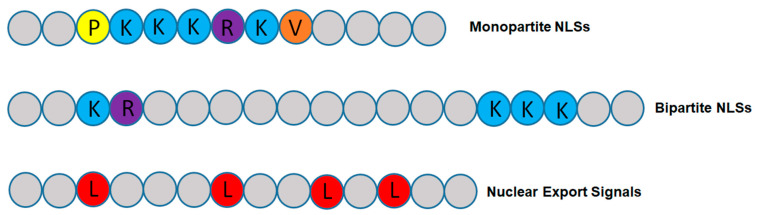
Conserved sequences of monopartite NLS, bipartite NLS and the mostly common form of NES. Each circle with different color is a different amino acid. P, proline; K, lysine; R, arginine; V, valine; L, leucine.

**Figure 4 ijms-22-04165-f004:**
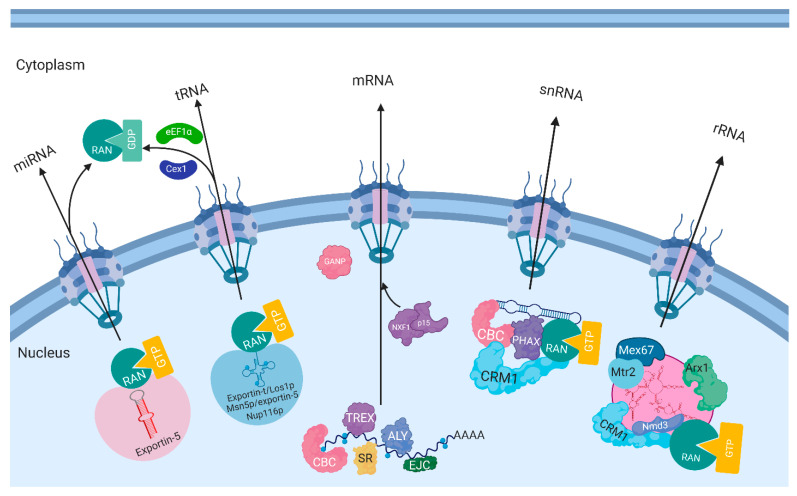
Nuclear export of different types of RNAs. RanGTP-RanGDP gradient provides driving force for nuclear export of miRNAs, tRNAs, snRNAs, and rRNAs, but not mRNAs. The export of mRNAs uses a non-karyopherin transport receptor and does not directly depend on the RanGTP-RanGDP gradient. The export receptors are exportin-5 for miRNAs, exportin-t for tRNAs, NXF for mRNAs, and CRM1 for snRNAs and rRNAs.

**Figure 5 ijms-22-04165-f005:**
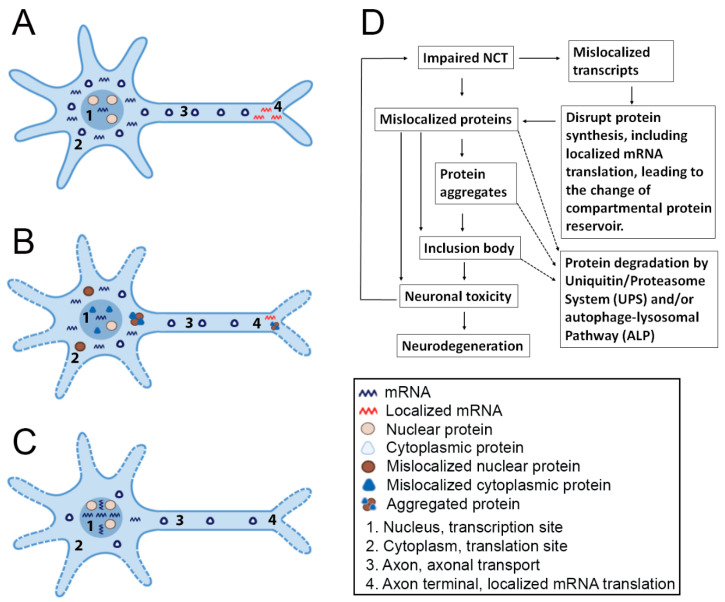
The linkages between impaired NCT and neurodegeneration. (**A**) A neuron possesses normal NCT activities. Transport mechanisms enable both protein and transcript cargos to reach and retain in appropriate compartments at subcellular locations of nucleus (1), cytoplasm (2), axons (3), and axon terminals (4), at which some localized mRNAs are translated into proteins. The majority of mRNAs localize in the cytoplasm. (**B**) A neuron with dysregulated protein NCT. The normal distributions of protein cargos are disrupted and cause protein mislocalization in either nucleus (1) or cytoplasm (2). Abnormal protein-protein interactions and protein aggregates (3 and 4) could disrupt neuronal functions and finally lead to neurodegeneration. (**C**) A neuron with impaired nuclear mRNA export. Nuclear mRNA accumulation and mislocalization (1) will occur and disrupt the protein synthesis, leading to the change of compartmental protein reservoir (2 and 3). Localized mRNA translation at axons and dendrites could be diminished (4). (**D**) Linkages between impaired NCT and neurodegeneration. Impaired NCT causes protein and transcript mislocalization, which will disrupt the homeostasis of local protein reservoirs and cause abnormal protein-protein interactions and/or protein aggregation, which may lead to neuronal toxicity, including interfering with NCT factors and signaling pathways, and finally causes neurodegeneration.

**Table 1 ijms-22-04165-t001:** Characteristics of human nucleoporins and their homologous in *Saccharomyces cerevisiae.*

Nups	AA	Structure	*S. cerevisiae*
Outer ring Nups			
Nup160	1436	Unstructured region, β-propeller domain, and α-helical solenoid	Nup120
Nup75	656	Domain invasion motif and α-helical solenoid	Nup85
Seh1	360	β-propeller domain	Seh1
Nup96	936	Unstructured region and α-helical solenoid	Nup145C
Sec13	322	β-propeller domain	Sec13
Nup107	925	Unstructured region and α-helical solenoid	Nup84
Nup133	1156	Unstructured region, β-propeller domain, and α-helical solenoid	Nup133
Nup43	380	β-propeller domain	-
Nup37	326	β-propeller domain	-
ELYS	2266	Unstructured region, β-propeller domain, and α-helical solenoid	-
Inner ring Nups			
Nup205	2012	α-helical solenoid	Nup192
Nup188	1749	α-helical solenoid	Nup188
Nup93	819	Unstructured region, Rec-A-like domain, and α-helical solenoid	Nic96
Nup155	1391	Unstructured region, β-propeller domain, and α-helical solenoid	Nup157 and Nup170
Nup53	326	Unstructured region, Rec-A-like domain, and RNA recognition motif	Nup53 and Nup59
Nup54	507	FG-repeat region and coiled-coil region	Nup57
Nup58	599	FG-repeat region and coiled-coil region	Nup49
Bridging complex			
Nup62	522	FG-repeat region and coiled-coil region	Nsp1
Nup98	880	FG-repeat region, Gle2-binding sequence, unstructured region, and auto-proteolystic domain	Nup100, Nup116, and Nup145N
Cytoplasmic filament			
Rae1	368	Unstructured region and β-propeller domain	Gle2
Nup42	423	FG-repeat region, Zinc finger region, and Gle2-binding sequence	Nup42
Nup88	741	Unstructured region, β-propeller domain, and coiled-coil region	Nup82
Nup214	2090	Unstructured region, β-propeller domain, coiled-coil region, and FG-repeat region	Nup159
DDX19	479	Unstructured region and Rec-A-like domain	Dbp5
Gle1	698	Unstructured region, coiled-coil region, and α-helical region	Gle1
Nup358	3224	α-helical solenoid, unstructured region, Zinc finger region, E3 ligase domain, Ran-binding domain, and cyclophilin domain	-
Nuclear basket			
Nup153	1475	Unstructured region, Zinc finger region, and FG-repeat region	Nup1
Nup50	468	Unstructured region, FG-repeat region and Ran-binding domain	Nup2
Tpr	2363	FG-repeat region and coiled-coil region	Mlp1 and Mlp2
Pore membrane proteins			
Ndc1	674	Unstructured region, transmembrane helices, Rec-A-like domain, and α-helical region	NDC1
NUP210	1886	Β-strand region, coiled-coil region, and unstructured region	-
POM121	1249	Coiled-coil region, and unstructured region	-
ALADIN	546	β-propeller domain	-

**Table 2 ijms-22-04165-t002:** Nomenclature and functions of human importins and exportins.

Human Kaps	Other Names	Functions	References
Importin-α	Kapα	Nuclear import of proteins containing either a simple or bipartite NLS motif.	[30]
Importin-β1	Kapβ1 Importin-90 NF-p97 PTAC97	Nuclear import of ribosomal proteins, H1 histone, HIV-1 Rev, SNAI1 and PRKCI; In vitro, nuclear import of other histones	[31,32,33,34,35]
Importin-4	Imp-4b RanBP4	Nuclear import of ribosaml protein, RPS3A;In vitro, nuclear import of human cytomegalovirus UL84 by recognizing a non-classical NLS	[36]
Importin-5	Kapβ3 Imp-β3 RanBP5	Nuclear import of ribosomal proteins and HIV-1 Rev and reverse transcription complex (RTC) integrase; In vitro, nuclear import of H2A, H2B, H3 and H4 histones; Nuclear import of CPEB3 after neuronal stimulation.	[31,32,37,38]
RanBP6		Act as a nuclear transport receptor	[39]
Importin-7	RanBP7	Nuclear import of ribosomal proteins and H1 histone; In vitro, nuclear import of other histones.	[31,32,37,38]
Importin-8	RanBP8	In vitro mediates the nuclear import of SRP19	[40]
Importin-9	RanBP9	Nuclear import of ribosomal proteins, actin, and histone H2A and H2B; Prevents the cytoplasmic aggregation of RPS7 and RPL18A by shielding exposed basic domains.	[36]
Importin-11	RanBP11	Nuclear import of UBE2E3, and of RPL12	[41]
Importin-13	Kap13 RanBP13	Nuclear import of UBC9, the RBM8A/MAGOH complex, PAX6 and probably other members of the paired homeobox family; Nuclear export of eIF-1A, and the cytoplasmic release of eIF-1A is triggered by the loading of import substrates onto IPO13.	[42,43]
Transportin-1	Kapβ2 Imp-β2	Nuclear import of ribosomal proteins ADAR/ADAR1 isoform 1 and isoform 5 in a RanGTP-dependent manner.; In vitro, nuclear import of H2A, H2B, H3 and H4 histones, and SRP19.	[31,38,40,44,45,46]
Transportin-2 Isoform 2	Kapβ2b	Nuclear export of mRNA and import of HuR.	[47,48,49]
Transportin-2		Same as Transportin-2 Isoform 2	[47,48]
Transportin-3	Trn-SR Imp-12	Nuclear import of splicing factor serine/arginine (SR) proteins, HIV-1 pre-integration complex (PIC)	[50,51,52,53,54,55,56,57,58,59,60]
Transportin-SR2	Isoform 2 of Trn-SR	Nuclear import of phosphorylated SR proteins	[52]
Exp-t	Xpo-t	Binds to RanGTP and cooperatively export mature tRNA (Figure 3)	[61]
Exportin-1	CRM1	Nuclear export of unspliced or incompletely spliced viral RNAs and proteins	[62,63,64,65,66,67]
Exportin-2	CAS CSE-1	Nuclear export of importin alpha	[68]
Exportin-4		Nuclear export of different protein cargos	[69,70]
RanBP17		Nuclear export activity by binding to the GTP and RanGTPase	[71]
Exportin-5	RANBP21	Nuclear export of proteins bearing a double-stranded RNA-binding domain (dsRBD) and double-stranded RNAs (cargos), isoform 5 of ADAR/ADAR1, micro-RNA precursors, synthetic short hairpin RNAs, deacylated and aminoacylated tRNAs, and adenovirus VA1 dsRNA.	[72,73,74,75,76]
Exportin-6	RANBP20	Nuclear export of actin and profilin-actin complexes in somatic cells.	[77]
Exportin-7	RANBP16	Nuclear export of different protein cargos	[71,78]

**Table 3 ijms-22-04165-t003:** Nuclear export of different types of RNAs.

RNAs Type	Key Factors	Functions in Nuclear RNA Export	References
mRNAs	NXF1/nxf1 (TAP/p15)	A transport receptor heterodimer. Bridges the interaction between mRNPs and FG Nups to facilitate transport of mRNPs through the NPC	[105,106]
	CRM1 (exportin 1)	Another major transport receptor, export a subset of endogenous mRNAs and HIV mRNA via adaptor proteins	[107]
tRNAs	Los1p/exportin-t, Msn5p/exportin-5, Nup116p	Export tRNA from the nucleus to the cytoplasm occurs through nucleopores, in an energy-dependent mechanism proceeds via the Ran pathway.	[108,109]
rRNAs	CRM1/Exp-t	rRNA export depend on Impβ family	[110]
snRNAs	Mex67 and Xpo1/Crm1	pre-snRNAs immediately exported into the cytoplasm upon binding of the export receptor Mex67-Mtr2 and the karyopherin Xpo1/Crm1	[111]
miRNAs	Exportin-5	Exportin-5 (Exp5) mediates efficient nuclear export of short miRNA precursors (pre-miRNAs)	[73]

**Table 4 ijms-22-04165-t004:** Neurodegenerative diseases with impaired NCT and possible mechanisms.

ND Diseases	Mutations	Related NCT Defect	Mechanisms
ALS/FTD	C9orf72 [139], SOD1 [140], FUS [141,142], TARDBP [143,144], CHCHD10 [145,146], UBQLN2 [147], SQSTM1 [148,149], OPTN [150,151], VCP [152,153], TBK1 [154], MAPT [155], GRN [156], CYLD [157]	Impairment of Nup35, Nup50, Nup54, Nup58, Nup62, Nup88, Nup93, Nup98, Nup107, Nup153, Nup155, Nup160, Nup188, Nup205, Nup214, Nup358, Kapβ1, RanGAP1, importin-α/β, Xpo5, Gle1, Nxf1 [158,159,160,161,162,163,164].	Cytoplasmic aggregates of DPRs, SOD1, TARDBP, FUS and tau [130,131,133,165,166,167].
HD	HTT [168]	Impairment of RanGAP1, Rae1, Nup Sec13, Nup62, Nup88, Gle1 [12,13,169]	Nuclear accumulation of mutant Huntingtin (mHTT), disrupts NE architecture and NPCs, sequesters NCT factors (Gle1 and Ran-GAP1) [12,13,134].
AD	APP [170], Presenilins 1 [171], Presenilins 2 [171], ABCA7 [172] and SORL1 [173]	Impairment of Nup62, Nup98, importin-α1, Accumulation of NTF2 [14,128,174]	Accumulation of amyloid plaques and neurofibrillary tangles [128,129,175]. Tau directly interacts with nucleoporins of NPCs and affects their structural and functional integrity [14].

ALS: Amyotrophic lateral sclerosis; FTD, Frontotemporal dementia; HD, Huntington’s disease; AD, Alzheimer disease.

## Data Availability

Not applicable.

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
