# Peer review of "Nucleocytoplasmic Transport: Regulatory Mechanisms and the Implications in Neurodegeneration"

_ijms, 2021, doi:10.3390/ijms22084165_

Round 1

Reviewer 1 Report

This review is definitely very interesting and at the moment there are not many other works on the nucleocytoplasmic transport defects and the correlation to neurodegenerative diseases, so I think it is to be published.
The only remark concerns the addition of a paragraph on Parkinson's disease, which is quoted many times, but no details are given.

Author Response

Re: We appreciate the reviewer’s positive comments on our manuscript.

Because many studies have been reported recently, we have taken ALS/FTD, HD and AD as examples in understanding how impaired nucleocytoplasmic transport (NCT) contributes to neurodegenerative diseases. Parkinson’s disease was not chosen as a detailed example because the limited length of the manuscript and very few studies about direct linkage between NCT and Parkinson’s disease (To our best knowledge, only one major publication available so far, Chen et al 2020. PMID: 32954426). With more studies published, Parkinson’s disease could be another good example in this topic in the future.

Reviewer 2 Report

The review manuscript by Ding et al. describes regulatory mechanisms of nucleocytoplasmic transport and its implications in neurodegeneration. 

This reviewer wants to congratulate the authors on a terrific job. The manuscript is informative and easy to read. 

Author Response

Re: We greatly appreciate the reviewer’s positive comments on our manuscript.

Reviewer 3 Report

This narrative review explains the role of nucleocytoplasmic transport in neurodegeneration. The article is interesting, it addresses a current topic. In my opinion, this work is acceptable.

  • The title and abstract reflect the content of the work.
  • The manuscript is concisely written, and the conclusions drawn are supported by the data and the adequate referencing of past studies.
  • The study is scientifically sound and advances further the knowledge in this very important area of neurodegeneration.

Author Response

Re: We greatly appreciate the reviewer’s positive comments and high evaluation about our manuscript.